# Demographic and professional risk factors of SARS-CoV-2 infections among physicians in low- and middle-income settings: Findings from a representative survey in two Brazilian states

**Giuliano Russo**[1]*, **Alex Cassenote**[2], **Bruno Luciano Carneiro Alves De Oliveira**[3], **Mário Scheffer**[2]

**1** Wolfson Institute of Population Health, Queen Mary University of London, London, United Kingdom, **2** Department of Preventive Medicine, University of São Paulo, São Paulo, São Paulo, Brazil, **3** Department of Medicine I, Federal University of Maranhão, São Luís do Maranhão, Brazil

* g.russo@qmul.ac.uk

## Abstract

Health workers (HWs) are a key resource for health systems worldwide, and have been affected heavily by the COVID-19 pandemic. Evidence is consolidating on incidence and drivers of infections, predominantly in high-income settings. It is however unclear what the risk factors may be for specific health professions, particularly in low- and middle-income countries (LMICs). We conducted a cross-sectional survey in a representative sample of 1,183 medical doctors registered with Brazil's Federal Council of Medicine in one developed (São Paulo) and one disadvantaged state (Maranhão). Between February-June 2021, we administered a telephone questionnaire to collect data on physicians' demographics, deployment to services, vaccination status, and self-reported COVID-19 infections. We performed descriptive, univariate, and multilevel clustered analysis to explore the association between physicians' infection rates, and their sociodemographic and employment characteristics. A generalized linear mixed model with a binomial distribution was used to estimate the adjusted odds ratio. We found that 35.8% of physicians in our sample declared having been infected with SARS-CoV-2 virus during the first year of the pandemic. The infection rate in Maranhão (49.2%) [95% CI 45.0–53.4] was almost twice that in São Paulo (24.1%) [95% CI 20.8–27.5]. Being a physician in Maranhão [95% CI 2.08–3.57], younger than 50 years [95% CI 1.41–2.89] and having worked in a COVID-19 ward [95% CI 1.28–2.27], were positively associated with the probability of infection. Conversely, working with diagnostic services [95% CI 0.53–0.96], in administrative functions [95% CI 0.42–0.80], or in teaching and research [95% CI 0.48–0.91] were negatively associated. Based on our data from Brazil, COVID-19 infections in LMICs may be more likely in health systems with lower physician-to-patient ratios, and younger doctors working in COVID-19 wards may be infected more frequently. Such findings may be used to identify policies to mitigate COVID-19 effects on HWs in LMICs.

**Data Availability Statement:** All relevant data are within the paper and its Supporting information files.

**Funding:** This study received support from the Confap-MRC call for Health Systems Research Networks. Specifically, GR received the award from the Newton Fund/ Medical Research Council (UK), Grant Reference MR/R022747/1. AC and MS received the award from the Fundação de Amparo à Pesquisa do Estado de São Paulo (FAPESP-Brazil), 2017/50356-7. BO received the award from the Fundação de Amparo à Pesquisa e ao Desenvolvimento Científico e Tecnológico do Maranhão (FAPEMA-Brazil), COOPI-00709/18. The funders had no role in study design, data collection and analysis, decision to publish, or preparation of the manuscript.

**Competing interests:** The authors have declared that no competing interests exist.

# 1 Introduction

Health workers (HWs) are a crucial healthcare resource, as they are one of the key health system's pillars, they deliver healthcare services, operate the health sector's building blocks [1], and in most health systems, represent the largest single item of expenditure [2, 3]. Their role and importance have been particularly evident during COVID-19, when they have been at the frontline of curative and preventive services, and led the clinical response to the pandemic [4]. Precisely because of their exposure, many HWs across the world contracted the virus, and some have died [5].

It is not clear how many HWs have been lost to the 2019 novel coronavirus disease (COVID-19), and what the key risk-factors may be. The World Health Organization estimates that between 80,000 and 180,000 may have actually died, if country-specific COVID-19 infection and fatality rates are applied to the 135 million-strong global health workforce [6]. There is evidence the associated burden of mental health disorders has been disproportionately high among HWs [7]. However, the available evidence is not conclusive on: (a) whether COVID-19 prevalence among HWs is necessarily higher than in the general population; (b) what specific health professional would be more at risk; (c) what the key risk factors and relevant exposure are, particularly for doctors in LMICs [8].

Cross-sectional, cohort, and hospital-based prevalence studies have been carried out to estimate COVID-19 infections among health workers. An observational cohort study from Denmark [9] screening for SARS-CoV-2 infections among medical, nursing and students personnel identified 4·04% seropositive health-care workers. A metanalysis of the prevalence of staff infections in hospital settings from 47 eligible studies in America, Europe and Asia [10], found the prevalence of infection was 7% for those studies using antibody tests, while for those studies using Polymerase Chain Reaction (PCR) tests, prevalence of infections was 11%. A highly referenced systematic review of the evidence from the first semester of the pandemic [11] estimated a total of 152,888 infections among health workers and 1,413 deaths worldwide.

A living review of the epidemiology of COVID-19 among HWs [12] concluded that these account for a significant proportion of global coronavirus infections worldwide; nurses may be the personnel most at risk, and severity of illness is lower in non-patient facing workers [13]. Professional exposures such as involvement in intubations, direct patient contact, or contact with bodily secretions, were associated with increased risk of infection. However, there is also evidence that private community exposure may be a stronger risk factor than work exposure [14].

A number of explanations have been suggested for HWs' higher rates of SARS-CoV-2 virus infections, from high occupational exposure and easier access to testing equipment, to lower positive practice towards COVID-19 [15]. Exposure to at-risk-patients from high-infection regions or from COVID-19 wards would be relevant risk factors [16], but it is not clear what specific healthcare settings increase the risk of infection, and who exactly are the most at risk 'frontline workers'. A study assessing COVID-19 hospital admission in Scotland [17] found that older, male HWs were more at risk of infection and hospitalization; patient-facing health-care workers and their households were at higher risk compared with non-patient facing ones. A prospective, observational cohort study in the UK and the US [18] concluded that self-reported infection rates among frontline health-care workers are higher than in the general population. But on the other hand, a study from a UK hospital [19] showed that infection levels were greater among HWs working in housekeeping, acute medicine, and general internal medicine, with surprisingly lower rates for those working in intensive care.

Some evidence is available from LMICs. An analysis of 101 medical staff admitted for COVID-19 in a Wuhan hospital in China [20] showed these to be younger than typical

patients, and displaying milder symptoms. A study from Iran [15] also found COVID-19 infections among HWs to be greater among younger (<35) professionals, particularly among women and nurses. Another study in 14 hospitals from Qatar [21] found HWs infections to be more frequent in non-COVID-19 facilities, where Personal Protective Equipment (PPE) would be only erratically used. Another investigation of COVID-19 cases among HWs in Oman [22] confirmed that the majority of these were among young (>45 years), female workers, predominantly deployed in primary care settings.

A study assessing COVID-19 infections among health workers in a Rio de Janeiro hospital in Brazil [23] found an overall seroprevalence of 30%. Non-white staff (mostly hospital support workers) with lower income and schooling, as well as users of the mass transportation system, showed the highest infection rates. HWs income level, schooling and work modality appeared as negative predictors. Analogous studies from São Paulo hospitals [24, 25] showed HWs infection rates to be similar to those in the wider population. Male and non-clinical workers appeared to be more at risk but working in COVID-19 services was not associated with higher levels of infections.

Despite these general studies, a knowledge gap exists on COVID-19 risk factors and on the association of personal and professional characteristics with SARS-CoV-2 virus infections for HWs in low-income settings, particularly for doctors. This is particularly of interest in South America which has been hit particularly hard by the epidemic [26]; an estimated 13,525 health workers died in Brazil, the world's second largest loss [6].

In Brazil, initial shortcomings in the availability of tests hampered the collection of epidemiological data for SARS-CoV-2 infections among health professionals. For the first year of the pandemic there was no official testing protocol for the wider population or HWs [27]. The Pan American Health Organization's guidelines in 2021 established Nucleic Acid Amplification Tests as the gold standard for population testing—in particular those based on Reverse Transcription Polymerase Chain Reaction (or RT-PCR) -, to be complemented in a later stage by Antigen Rapid Detection tests (Ag-RDT or lateral flow test), to achieve the highest possible coverage among the wider population [28]. Testing asymptomatic cases was only recommended for key workers subject to increased exposure, including HWs. After testing positive, cases were asked to isolate for 10 days. However, given the persistent shortages of COVID-19 tests, there are reports that any available testing method was used for HWs, including clinical diagnosis by another qualified health professional [29].

In this paper we report the results of a study to explore the self-reported SARS-CoV-2 infection rates among physicians in Brazil, with the objective to identify the risk factors associated with infections. The study aimed at contributing to the existing knowledge on impacts of COVID-19 on health workers in LMICs, providing an evidence base for local and international policies to mitigate effects of the pandemic.

## 2 Methods

### 2.1 Ethics statement

This study received the approval from the Research Ethics Committees of the Federal University of Maranhão (CEP UFMA 3.051.875), and from the Faculty of Medicine of the University of São Paulo (CEP FMUSP 3.136.269), and was approved by Brazil's Federal Council of Medicine, that provided the list of physicians registered with the council of medicines in the two states, and the respective telephone contact details.

All the physicians contacted via telephone were informed beforehand of the objectives of the survey; their consent was verbally obtained to participate in the survey, and recorded by the data collector as part of the database entry. All the forms and questionnaires were

anonymized through a double coding system organized by the Faculty of Medicine of the University of São Paulo. Such process for obtaining consent and guaranteeing anonymity was approved by the Institutional Review Boards of the University of São Paulo and Federal University of Maranhão.

## 2.2 Study design

As part of a wider research project on the impact of COVID-19 and the associated economic crisis on Brazil's health system [30], we conducted a representative cross-sectional telephone survey among registered physician during the second year of the pandemic, in two states, São Paulo and Maranhão.

These two states represent extreme cases of economic and health system disparities [31]. With almost 47 million people, São Paulo has one of Brazil's highest income per capita (US$ 12,776), and 43% of its population is covered by private health insurance schemes [32]. Conversely, Maranhão has slightly more than 7 million people, its income is about one third that of São Paulo's, and just 7% of the population own a private health plan (Table 1).

São Paulo has three times the number of physicians per capita as Maranhão [33], and appears to have been hit harder by the pandemic [34], with three times the number of COVID-19 deaths per capita than in the other state, and twice the number of infections among its population (Table 1).

## 2.3 Data collection and sampling strategy

The national physician database for the two states was provided by Brazil's Federal Council of Medicine. The survey sample was calculated by the Faculty of Medicine of the University of São Paulo, and the actual survey was carried out by the survey services institute 'Datafolha', under the technical supervision of the academic partners of the study.

A representative cross-sectional study including 1,183 physicians was conducted in 2021. Because of the substantial difference in the physician population in the two settings, two independent sample sizes were calculated (one per state) based on a total of 152,511 active medical registries in São Paulo (N = 144,852) and Maranhão (N = 7,659) from the Federal Medical Council Medicine database (*Conselho Federal de Medicina—CFM*), using a 95% confidence level with 5% margin of error and statistical power of 80% (see S1 Text). Proportional stratified sampling was constructed following the physicians' distribution for gender, age, state and local of address (capital or countryside).

Substitution was carried out in cases of unsuccessful contact or refusal to participate in our survey; 1,183 physicians were randomly selected, and five substitutions were identified for each sampled physician. Substitution sampling followed the same stratification criteria used

**Table 1. Selected economic, health system, and COVID-19 indicators for the study locations at the time of the study.**

| Area | Population (2021)* | GDP per capita (USD 2019)* | Phys/100,000** | COVID-19 Cases/100,000*** | COVID-19 Deaths/100,000*** | COVID-19 fatality*** |
|---|---|---|---|---|---|---|
| Brazil | 213,317,639 | 8,924 | 251.3 | 12,815.6 | 298.2 | 2.2% |
| São Paulo | 46,649,132 | 12,776 | 310.5 | 9,345.2 | 319.1 | 3.4% |
| Maranhão | 7,153,262 | 3,453 | 107.1 | 5,030.3 | 143.4 | 2.9% |

Source:

*IBGE, 2022;

** Brazilian Medical Council, 2021;

*** CONASS, 2021

for the initial sample calculation. We controlled sample replacements for state, sex, and age, so that every physician who did not agree to participate was replaced by an individual with the same gender, age, and specialty characteristics to avoid selection bias.

Primary data were collected via a telephone survey carried out by 8 data collectors, including one field coordinator, 6 experienced interviewers, and two administrative staff responsible for checking missing data. Sample size calculations, sample selection, questionnaire design, substitution control, database assembly and data analysis were performed by the authors of this papers. Three senior researchers from the medical demography field previously piloted and calibrated the questionnaire with 30 interviewees to estimate the substitution rate. Reproducibility was tested by sampling a random sample after the field collection and repeating the interview, resulting in 100% agreement. All in all, 8,132 physicians were contacted in order to secure the target 1,183 valid interviews. The physicians who started but did not complete the interview were 1,222 (2,445–1,183). Those who were busy, engaged, or asked to be called later were 1,423. No systematic difference was found in personal characteristics or employment among those who declined to participate.

Data collection was carried out between the 16[th] February and 15[th] June 2021 by the Datafolha Research Institute under supervision of the authors' research institutions. The interviews consisted of a 30-minute telephone questionnaire, containing 30 questions ranging from multiple, closed questions to interdependently concatenated and semi-opened questions (see S2 Text).

Patient and public involvement. Medical doctors as well as members of the public were consulted and participated in the design of the original version of the survey questionnaire. The questionnaire was then piloted by Datafolha in a subset of ten doctors in the two states, and a final version was elaborated following the feedback received.

## 2.4 Data analysis

Self-reported infection to the SARS-CoV-2 virus during the past year was the selected outcome variable, distinguishing between the severity of the infection events (asymptomatic, mild, and severe). We used sociodemographic characteristics (sex, gender, age, and income), medical employment characteristics (such as medical specialty, training and years of service), and type of work carried out in COVID-19 and regular wards (intensive care, inpatient or outpatient care, distance-based consultation, or research), as independent, explanatory variables.

We performed descriptive, univariate, and multilevel clustered analysis on the dataset. A generalized linear mixed model with a binomial distribution was used to estimate the adjusted odds ratio [35] as these allow for the inclusion of random or cluster-specific effects in the linear predictor. The inclusion of random effects in the linear predictor reflects the idea that there is natural heterogeneity across geographic clusters in their regression coefficients. Such method has been used extensively in health services research [36]. The 'enter' method was used for the selection of variables (those reported in Table 3), that were included all at the same time. ANOVA tests were employed to verify the equality hypothesis among the different models. Data were shown as absolute frequency and proportion with a 95% confidence interval. The adjustment of different models was verified by indicators of residual deviance and the Akaike information criterion (AIC) [37]. The database developed by the Datafolha data collectors was exported to the Statistical Package for the Social Sciences (SPSS) version 26 for Windows (International Business Machines Corp, New York, USA) and R-GUI version 3.5.3 [38] for statistical analysis. All the significance levels were set to $p < 0.05$.

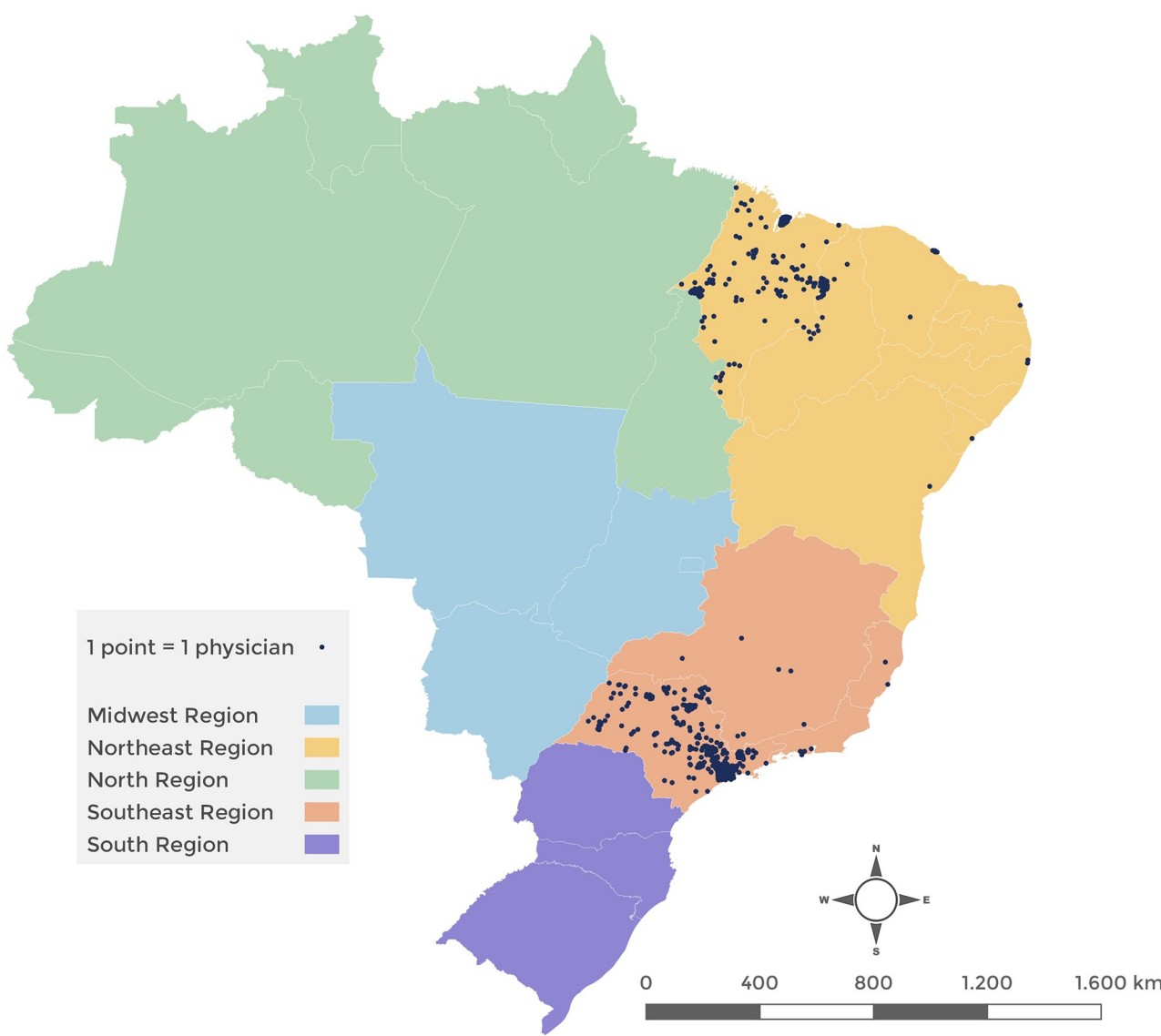

**Fig 1. Geographical location of the surveyed physicians in São Paulo and Maranhão.** Note to Fig 1: Points outside the states of São Paulo and Maranhão refer to registered physicians who have recently retired or are active in more than one state. The base layer map of Brazil is from the Brazilian Institute of Geography and Statistics. Link: https://www.ibge.gov.br/geociencias/organizacao-do-territorio/malhas-territoriais/15774-malhas.html?=&t=downloads.

## 3 Results

Our sample included 1,183 physicians, 551 from Maranhão and 632 from São Paulo, approximately split equally between urban and rural areas (see the graphical map of physicians" locations in Fig 1 below).

Most physicians in our survey (61.6%) worked both in public and private sector jobs (dual practice), with only 25.4% of them employed exclusively in the public. Most of our physicians declared engaging in outpatient care (82.7%) in hospital or clinics settings, and 63.4% of them were deployed directly to the delivery of COVID-19 services, to COVID-19 wards or to COVID-19-specific outpatient care. Engagement in the delivery of specific healthcare services

**Table 2. Characteristics of physicians in the survey sample.**

| Characteristics/variables | n | Proportions and Confidence Intervals |
|---|---|---|
| Gender | | |
| Male | 665 | 56.2% (53.4%-59.0%) |
| Female | 518 | 43.8% (41.0%-46.6%) |
| Age | | |
| < 35 yeas | 404 | 34.2% (31.5%-36.9%) |
| 35 to 50 years | 405 | 34.2% (31.6%-37.0%) |
| > 50 years | 374 | 31.6% (29.0%-34.3%) |
| State | | |
| Maranhão (MA) | 551 | 46.6% (43.7%-49.4%) |
| São Paulo (SP) | 632 | 53.4% (50.6%-56.3%) |
| Geographical location of deployment | | |
| Rural areas (Interior) | 598 | 50.5% (47.7%-53.4%) |
| Urban areas around capital cities | 585 | 49.5% (46.6%-52.3%) |
| Health sector of deployment | | |
| Exclusively Private | 153 | 12.9% (11.1%-14.9%) |
| Exclusively Public | 301 | 25.4% (23.0%-28.0%) |
| Dual practice | 729 | 61.6% (58.8%-64.4%) |
| Employment in specific health services | | |
| Outpatient clinical services (hospital or clinics) | 978 | 82.7% (80.4%-84.7%) |
| Diagnostic tests equipment-related services | 392 | 33.1% (30.5%-35.9%) |
| Surgery (in-patient care) | 459 | 38.8% (36.1%-41.6%) |
| Outpatient surgery | 450 | 38.0% (35.3%-40.8%) |
| Administrative position | 288 | 24.3% (22.0%-26.9%) |
| Teaching and research | 312 | 26.4% (23.9%-28.9%) |
| Engagement with COVID-19 services | | |
| No | 433 | 36.6% (33.9%-39.4%) |
| Currently working with COVID-19 services | 631 | 53.3% (50.5%-56.2%) |
| Worked in the past, but not currently | 119 | 10.1% (8.4%-11.9%) |
| Type of COVID-19 services delivered | | |
| COVID-19 ward or Intensive care unit (ICU) | 524 | 44.3% (41.5%-47.2%) |
| Outpatient COVID-19 care | 490 | 41.5% (38.7%-44.3%) |
| Telemedicine or other distance-based COVID-19 services | 178 | 15.1% (13.1%-17.2%) |
| Research on COVID-19 | 61 | 5.2% (4.0%-6.5%) |
| Epidemiological surveillance or COVID-19 boards | 53 | 4.5% (3.4%-5.8%) |

was non-exclusive, as many physicians declared to work concomitantly in multiple wards, services, and sectors (Table 2). No significant difference was found in physicians' characteristics or employment across the two states.

At the time of administration of the survey, the vast majority of physicians declared having already been vaccinated (93%), with non-significant differences in the vaccine uptake among Maranhão and São Paulo physicians.

35.8% of all the physicians declared having been infected with SARS-CoV-2 in the previous year. Almost half (49.2%) [CI 45.0–53.4] of Maranhão physicians were infected, with the majority of them having suffered only mild or no symptoms (42.3%) (Fig 2). Conversely, in São Paulo 24.1% of physicians [CI 20.8–27.5] declared a SARS-CoV-2 infection, again with the majority recalling mild or no symptoms. In our sample, 4.5% of physicians were affected by

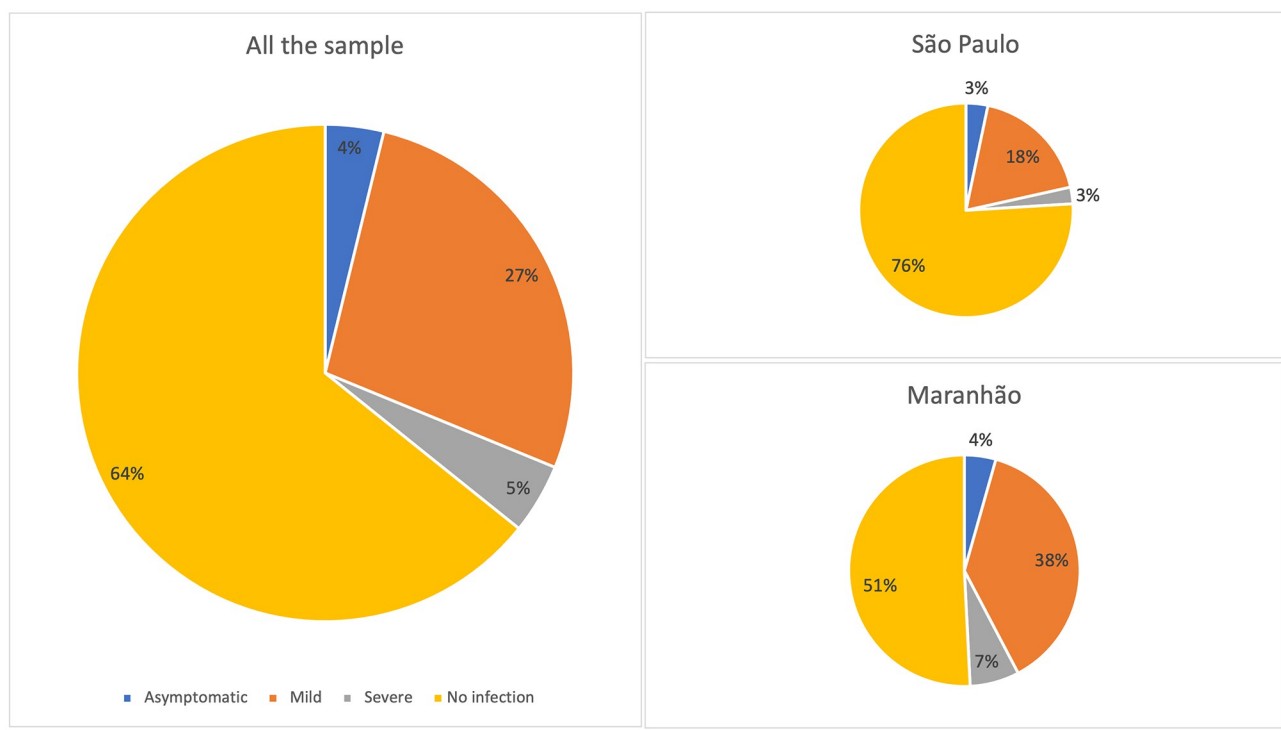

**Fig 2. Physician infection rates in the sample locations.**

severe COVID-19 symptoms, with the proportion in Maranhão (6.9%) [CI 5.0–9.2] significantly higher than in São Paulo (2.5%) [CI 1.5–4.0].

Such infections directly resulted in the loss of 7,117 workdays, representing 2.36% of the total working days in our physician sample.

The multivariate analysis showed that being from Maranhão State [95% CI 2.08–3.57], younger than 50 years [95% CI 1.41–2.89], and having worked in a COVID-19 ward [95% CI 1.28–2.27], were positively associated with the odds of being infected with the SARS-Cov-2 virus (Table 3). Conversely, working with diagnostic services such as X-rays or scans [95% CI 0.53–0.96], in administrative functions [95% CI 0.42–0.80], or in teaching and research [95% CI 0.48–0.91] had a protective effect on the probability of infection.

When multivariate models were run separately for each state, the results were not materially different, except for 'Working in diagnostic services' that was no longer significant for the São Paulo cohort (see S1 Table).

With specific reference to 'Age', when using '>50 years' as reference category, both the '35–50 years' and '35–50 year' categories showed strong, significant association with the outcome variable of likelihood of SARS-Cov-2 infections, although the latter appeared to have higher significance (p<0.001) and coefficient (B = 0.706).

On the association between physicians' engagement in specific services and probability of getting infected, only being deployed in COVID-19 hospital wards or ICU displayed a strong positive association for the two states, while deployment in administrative services displayed a strong negative significance. The association with 'being deployed in diagnostic services' was only significant for the Maranhão cohort, while 'deployment in teaching and research activities' was only significant for the São Paulo cohort, and lost significance when analysed jointly across the sample (see S1 Data).

**Table 3. Multivariate regression model for physicians personal and professional characteristics and probability of being infected with SARS-CoV-2 virus.**

|  | B | S.E. | OR (95%CI) | p-value |
|---|---|---|---|---|
| Global Model (n = 1,183) |  |  |  |  |
| Maranhão state (n = 551) | 1.004 | 0.138 | 2.72(2.08–3.57) | <0.001 |
| Gender Female (n = 518) | -0.117 | 0.138 | 0.88(0.67–1.16) | 0.396 |
| Age |  |  |  |  |
| > 50 years (n = 374) |  |  | Ref. | Ref. |
| 35 to 50 years (n = 405) | 0.706 | 0.182 | 2.02(1.41–2.89) | <0.001 |
| < 35 years (n = 404) | 0.577 | 0.173 | 1.78(1.27–2.50) | 0.001 |
| Employment in specific health services |  |  |  |  |
| Outpatient clinical services (hospital or clinics) (n = 978) | -0.289 | 0.176 | 0.74(0.53–1.05) | 0.102 |
| Diagnostic tests equipment-related services (n = 392) | -0.330 | 0.147 | 0.71(0.53–0.96) | 0.025 |
| Surgery (in-patient care) (n = 459) | 0.286 | 0.155 | 1.33(0.98–1.80) | 0.065 |
| Outpatient surgery (n = 450) | 0.047 | 0.155 | 1.04(0.77–1.42) | 0.760 |
| Administrative position (n = 288) | -0.533 | 0.160 | 0.58(0.42–0.80) | 0.001 |
| Teaching and research (n = 312) | -0.407 | 0.162 | 0.66(0.48–0.91) | 0.012 |
| Health sector |  |  |  |  |
| Exclusively Private (n = 153) |  |  | Ref. | Ref. |
| Exclusively Public (n = 301) | 0.248 | 0.256 | 1.28(0.77–2.11) | 0.333 |
| Dual practice (n = 729) | 0.129 | 0.236 | 1.13(0.71–1.80) | 0.585 |
| Type of COVID-19 services delivered |  |  |  |  |
| COVID-19 ward or Intensive care unit (ICU) (n = 524) | 0.537 | 0.146 | 1.71(1.28–2.27) | <0.001 |
| Outpatient COVID-19 care (n = 490) | -0.237 | 0.150 | 0.78(0.58–1.05) | 0.114 |
| Telemedicine or other distance-based COVID-19 services (n = 178) | 0.056 | 0.196 | 1.05(0.72–1.55) | 0.776 |
| Research on COVID-19 (n = 61) | -0.226 | 0.312 | 0.79(0.43–1.47) | 0.470 |
| Epidemiological surveillance or COVID-19 boards (n = 53) | 0.118 | 0.333 | 1.12(0.58–2.16) | 0.723 |

## 4 Discussion

Our survey found that 35.8% of physicians in two Brazilian states were infected with SARS-CoV-2 virus during the first year of the pandemic. Although most of them only experienced mild symptoms, 7,117 workdays were lost following the infections. The rate of infections varied considerably between Maranhão and São Paulo physicians, with the former being affected twice as much than the latter. Being a Maranhão physician, being younger than 50 years old, and deployed to a COVID-19 ward, were the factors positively associated with infections. Conversely, being deployed to diagnostic services, administrative functions, or to teaching and research, were found to have a protective effect. Physician gender, sector of employment, and deployment to other frontline services, were not significantly associated with SARS-Cov-2 infections.

These findings need to be interpreted with a degree of caution. Our survey sample did not include those physicians who died or were critically incapacitated because of COVID-19, therefore possibly underestimating overall infections. However, in absence of widespread SARS-Cov-2 virus testing—as it is often the case in LMICs -, numerous epidemiological studies have been conducted and published based on self-reported COVID-19 status [39–41]. Our data on infections, personal and professional characteristics were based on physicians' responses, which could have been affected by recall bias [42]. Shortcoming in the available tests and in Brazil's testing programme may imply that not all the HWs infected with SARS-CoV-2 virus may have reported correctly their status in the survey [43]. We did not collect information on availability and use of PPE among our physicians, which is considered in the

COVID-19 literature as a risk factor [18]. Finally, Maranhão and São Paulo represent very particular settings in terms of income distribution, organisation of healthcare services, and labour markets characteristics [44]; therefore, our study findings may not be entirely generalisable to the rest of Brazil, let alone to other low- and middle-income countries. Despite these limitations, a few conclusions can be safely drawn from our work.

We show that over a third of the medical workforce in Maranhão and São Paulo was infected with SARS-Cov-2 virus in the first year of the pandemic, with a substantial loss of labour. This is consistent with the findings from smaller studies from Brazil [23] and other LMICs [20–22], and therefore particularly relevant for those countries with a scarcity of healthcare resources, which will have been hit already particularly hard by the pandemic [45].

The higher infection rate among Maranhão physicians contrasted with lower population infection rates in the two states (Table 1). Our multivariate analysis confirmed that working in Maranhão was one of the most significant risk factors of physician infections in our cohort. The lower ratio of physicians per capita in Maranhão (1.1 per 1,000 in Maranhão Vs 3.2 per 1,000 in São Paulo) [33] may be a factor here, as during health emergencies a smaller workforce will necessarily engage in multiple functions and tasks across sectors, therefore increasing opportunities for infection. This is consistent with previous work [32] showing the differential impact of health system crises on unequal states in LMICs. If confirmed, such finding would be relevant for those studies forecasting effects of the pandemic on health workforces in different parts of the world [6]

Younger age was associated with higher infection rates among physicians in both Brazilian states, which, to some extent, contradicts the existing evidence on COVID-19 risk factors from high-income countries [46]. As our sample did not include those physicians who died of COVID-19 or were too ill to participate, a possible explanation is that some of the more vulnerable physicians were somewhat underrepresented in our study. However, since we stratified our sample by age group, it is unlikely that younger physicians were overrepresented in our study (the proportion of all age groups in our sample is similar to that of the universe of physicians in Brazil). Furthermore, our finding on younger doctors' higher rate of self-declared infections is consistent with the evidence from other LMICs [15, 20–22]. As new public and private medical schools are supplying the national workforce in Brazil [47], newly graduated physicians were predominantly deployed to COVID-19 wards and Nightingale hospitals in São Paulo and Maranhão, and older physicians were excluded from COVID-19 functions because considered more at risk [48, 49]. While such a policy may make sense from a health management and epidemiological point of view, it also poses ethical and equity questions on medical employment in LMICs [50, 51], particularly at a time when contracts of newly graduated physicians suffer for increasing casualisation in Brazil and worldwide [52, 53].

Not unexpectedly, our analysis shows that working in COVID-19 wards is associated with higher rates of infections among physicians, compared to working in administrative functions, teaching and research. Although this is consistent with what is already known on SARS-Cov-2 risk factors for health workers [12, 18], it is however surprising that other 'frontline' functions (such as COVID-19 outpatient visits or diagnostic services) were not significantly associated with increased odds of infections. Such a lack of effect could be of course explained by the suspension of some 'non-essential' services during the pandemic, which would have spared physicians from dangerous exposure [54]. But it is also likely that the definition of what constitute 'frontline health workers may not be that clear-cut, particularly in LMICs settings during an epidemic, where boundaries and functions become blurred, and health workers end up carrying out whatever functions are needed.

It was also surprising to see similar odds of infections among public and private sectors physicians, particularly as publicly funded health systems are believed to have borne the brunt

of the COVID-19 pandemic [55, 56], particularly in Brazil [57]. In the case of Brazilian's physicians it is more likely that blurred boundaries between public and private employment may have made it difficult to accurately distinguish between physicians of the Unified National Healthcare System (SUS), from those of the private sector. As previous research has shown that the majority of Brazilian doctors simultaneously engage in concomitant public and private forms of employment [58], and that private organisations often provide services within SUS [44], it is likely that the majority of physicians in our sample carried out functions simultaneously in public and private sector institutions, making it very difficult to identify the individual effects of the pandemic on either sectors.

If confirmed, the findings from this study have policy implications for the ongoing efforts to estimate the effects of the pandemic on HWs, as well as for health policymakers in LMICs. Emerging evidence from different countries hints that the WHO's high-end figure [6] of 180,000 COVID-19 deaths among HWs is probably an overestimation. Rather than applying population-based infection and mortality rates, our study suggests that rates among HWs may vary according to HWs density, with understaffed health systems and services suffering more infections and deaths. This could help refine projections, and re-focus policies to mitigate the impact of the pandemic on health workforces in less staffed settings.

Although medical doctors might not be the HWs most at risk of SARS-CoV-2 infections [13], our study suggests that COVID-19's impact on such key and expensive human resource is not negligible, and less staffed parts of a country's health systems may be more at risk. Health authorities worldwide should therefore give priority during epidemics to protect those services with the lowest HW to population ratios, knowing that risk for infections will be greater where workers are more scarce [55].

Finally, our work appears to confirm that younger doctors in LMICs may be more at risk of infection than their counterparts in HICs [15, 22], possibly because the health services in those countries have drawn from younger recruits to staff COVID-19 services. While such decisions can be justified in the light of younger doctors' less severe hospitalization and fatality rates, health authorities should make sure younger doctors are compensated through better contracts, and access to more secure parts of the market for physician services [59].

## 5 Conclusions

Health workers are essential resources for any health system, and they have borne the brunt of providing life-saving services during the COVID-19 epidemic. Knowledge gaps exist on SARS-CoV-2 virus infections, mortality and risk factors among HWs, particularly in LMICs. We conducted a cross-sectional telephone survey in a representative sample of physicians in the populous São Paulo and the disadvantaged Maranhão state in Brazil, with a view to identify the associated risk factors.

We found that more than a third of physicians in the two states were infected with SARS-CoV-2 virus in the first year of the pandemic. Although most of them only experienced mild symptoms, a substantial number of workdays were lost following the infections. The rate of infections varied considerably between Maranhão and São Paulo physicians, with the former being affected twice as much than the latter. Being a Maranhão physician, younger than 50 years, and deployed to a COVID-19 ward, were positively associated with infections. Conversely, being deployed to diagnostic services, administrative functions, or to teaching and research, were found to have a protective effect.

More research is needed to explore depth and nuances of the impact of the pandemic on HWs in LMICs. Our findings on the greater impact for physicians from less-staffed parts of

Brazil's health system carry implications for the identification of policy to mitigate COVID-19 effects on health workforces, and for their measurement worldwide.

## Supporting information

**S1 Text. Sampling equations for sampling replacement.**
(DOCX)

**S2 Text. Survey questionnaire.**
(DOCX)

**S1 Table. Maranhão and São Paulo regression models.**
(DOCX)

**S1 Data. Anonymised survey database.**
(XLSX)

## Acknowledgments

We are indebted to Prof. Trevor Sheldon and Dr. James Buchan for their valuable comments and suggestions to an early version of the manuscript.

## Author Contributions

**Conceptualization:** Giuliano Russo, Bruno Luciano Carneiro Alves De Oliveira, Mário Scheffer.

**Data curation:** Alex Cassenote, Bruno Luciano Carneiro Alves De Oliveira.

**Formal analysis:** Giuliano Russo, Alex Cassenote, Bruno Luciano Carneiro Alves De Oliveira.

**Funding acquisition:** Giuliano Russo, Mário Scheffer.

**Investigation:** Giuliano Russo, Alex Cassenote, Mário Scheffer.

**Methodology:** Giuliano Russo, Bruno Luciano Carneiro Alves De Oliveira.

**Software:** Alex Cassenote.

**Validation:** Alex Cassenote, Mário Scheffer.

**Writing – original draft:** Giuliano Russo, Alex Cassenote.

**Writing – review & editing:** Giuliano Russo, Bruno Luciano Carneiro Alves De Oliveira, Mário Scheffer.

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
