## [Decision Letter · Decision Letter 0]

14 Jul 2022

PGPH-D-22-00875

Demographic and professional risk factors of COVID-19 infections among physicians in low- and middle-income settings; findings from a representative survey in two Brazilian states.

Dear Dr. Russo,

Thank you for submitting your manuscript to PLOS Global Public Health. After careful consideration, we feel that it has merit but does not fully meet PLOS Global Public Health’s publication criteria as it currently stands. Therefore, we invite you to submit a revised version of the manuscript that addresses the points raised during the review process.

Please pay particular attention to Reviewer #2's comments on the research question and whether this study can truly answer the question asked. Please offer a comprehensive and detailed response to all comments raised upon revision.

We look forward to receiving your revised manuscript.

Kind regards,

Julia Robinson

Staff Editor

Journal Requirements:

1. In the ethics statement in the Methods, you have specified that verbal consent was obtained. Please provide additional details regarding how this consent was documented and witnessed, and state whether this was approved by the IRB.

2. Please amend your detailed online Financial Disclosure statement. This is published with the article. It must therefore be completed in full sentences and contain the exact wording you wish to be published.

State what role the funders took in the study. If the funders had no role in your study, please state: “The funders had no role in study design, data collection and analysis, decision to publish, or preparation of the manuscript.”

3. Please update your online Competing Interests statement. If you have no competing interests to declare, please state: “The authors have declared that no competing interests exist.”

4. Please provide separate figure files in .tif or .eps format and ensure that all files are under our size limit of 10MB.

5. Please add a full list of legends for all your Supporting Information files after the references list.

Additional Editor Comments (if provided):

Reviewers' comments:

Reviewer's Responses to Questions

**Comments to the Author**

1. Does this manuscript meet PLOS Global Public Health’s publication criteria? Is the manuscript technically sound, and do the data support the conclusions? The manuscript must describe methodologically and ethically rigorous research with conclusions that are appropriately drawn based on the data presented.

Reviewer #1: Yes

Reviewer #2: Partly

2. Has the statistical analysis been performed appropriately and rigorously?

Reviewer #1: Yes

Reviewer #2: Yes

3. Have the authors made all data underlying the findings in their manuscript fully available (please refer to the Data Availability Statement at the start of the manuscript PDF file)?

Reviewer #1: Yes

Reviewer #2: No

4. Is the manuscript presented in an intelligible fashion and written in standard English?

Reviewer #1: Yes

Reviewer #2: Yes

5. Review Comments to the Author

Reviewer #1: The manuscript reports an interesting subject, although already known, but very relevant to the current situations, where HCWs were working with increased exposure to the virus. It will be of keen interest to readers. The findings are in support of some studies which adds value due to the different demographics and local public health policies. The manuscript is well written but can be improved. My comments are as follows:

The manuscript should be edited regarding many punctuational and grammatical errors.

Abbreviations should be mentioned in full the first time they appear in text, such as COVID-19, SARS, etc.

SARS CoV-2 is the virus that causes infection, and COVID-19 is the disease. Proper usage of these terminologies is mandatory and should be corrected throughout the manuscript.

the authors should describe the method of COVID-19 detection extensively. Who were PCR performed among and what were the protocols? How were data obtained. What was the screening method among HCWs? Also, the limitation of the detection method should be mentioned in the manuscript and the reference [COVID-19: clinical or laboratory diagnosis? A matter of debate. doi: 10.1177/0049475520945446]

How was the multivariate analysis performed and what factors were entered into the analysis? More details in this regard are warranted.

is the sample skewed towards a particular subset of the overall population? These are issues that need to be addressed in the limitations in your discussion.

I recommend reporting the local countries protocols during the timeline of your study so that international readers could have an insight on the situation and applied protective measures during the study period.

I advise the authors to provide the geographical map of towns representing the divisions covered by this study. Pictorial representation of data gives clarity about study settings and epidemiological data of particular regions.

Describe reasons for high infection rates among HCW is your study (such as High occupational exposure, Easy access to testing equipment (PCR, CT), knowledge and awareness of symptoms therefore fast evaluation: HCWs have higher knowledge about COVID-19 compared to non-HCWs; However, lower positive practice towards COVID-19. (Erfani A, et al. Knowledge, attitude and practice toward the novel coronavirus (COVID-19) outbreak: a population-based survey in Iran. Bull World Health Organ. 2020;30(10.2471).)

and also a recent study regarding high post-infection protection after COVID-19 among healthcare workers (Sabetian G, et al. High Post-infection Protection after COVID-19 Among Healthcare Workers: A Population-Level Observational Study Regarding SARS-CoV-2 Reinfection, Reactivation, and Re-positivity and its Severity.)

In terms of depression, being a healthcare worker was an associated risk factor. (Shahriarirad, R., et al. The mental health impact of COVID-19 outbreak: a Nationwide Survey in Iran. Int J Ment Health Syst 15, 19 (2021). https://doi.org/10.1186/s13033-021-00445-3)

For consideration; With the large data set, there could be added value by potentially splitting the previously infected HCW into vaccinated and unvaccinated to retrospectively analyse the added protectivity of having a vaccine dose and prior SARS-CoV 2 infection, as explored by studies below

a. Demonbreun, A.R., Sancilio, A., Velez, et al, 2021. Comparison of IgG and neutralizing antibody responses after one or two doses of COVID-19 mRNA vaccine in previously infected and uninfected individuals.. EClinicalMedicine 38, 101018.. doi:10.1016/j.eclinm.2021.101018

b. Wang, Z., Muecksch, F., Schaefer-Babajew, D. et al. Naturally enhanced neutralizing breadth against SARS-CoV-2 one year after infection. Nature 595, 426-431 (2021). https://doi.org/10.1038/s41586-021-03696-9

• Many of the references are incomplete (e.g., no page or volume number) while the journal names are abbreviated in some while full in others. Please adjust based on the journal’s guideline

Reviewer #2: Title: Accurately depicts the content of this study

Abstract:

Introduction:

A very powerful opening statement. Could the authors find more up to date references to back up the statement of healthcare staff being the most important health expenditure (I believe the authors wish to state that healthcare spending by states/private firms is largely directed towards labour)? Also, what do the authors mean by health system building blocks? From what they’ve written, it does not seem to be the WHO health system building blocks.

Also I note that the references cited by the authors offer differing risk factors. For example, one reference highlights that nurses in psychiatry wards and community doctors have been more likely to become infected and die, and another reference states that cleaners are at greater risk than ICU doctors. However, another reference states that those working in aerosol generating jobs are at increased risk of exposure. Both statements can be true (i.e., better protective equipment may be available to those at most risk, or they may be more aware of infection control measures). However, the authors would improve the introduction by discussing in more detail why disparities exist within the evidence base that currently exists. This is especially important given the authors are grouping disparate countries such as China and Qatar together when discussing evidence in LMICs, and the heterogeneity in evidence should be viewed in light of the context it is from.

The 7th and 8th paragraph of the introduction seem to contradict each other. The 7th describes risk factors for COVID-19 infection among healthcare workers in Brazil, whilst the 8th paragraph denies this evidence exists. The authors position might be strengthened if they state they want to corroborate existing evidence, or if they explain that circumstances have changed since the original studies in Brazil were published.

Methods:

What was the sample size calculated? Were 1183 physicians sufficient?

How were the physicians randomly selected? Describe the methodology.

How many physicians initially identified refused to participate? How did you ensure there was no selection bias in the substitutes, as you choose to replace the physicians who refused to participate?

How did you assess the reliability and validity of the survey?

Detailed analysis plan that all sounds appropriate

Results:

Were there any differences in demographic characteristics between the physicians in the two different locations?

Results can be understood, but could be displayed in a more readable manner.

I’d like to know how each factor affected the outcome on a univariate model in addition to their effect in a multivariable model.

Discussion

A lot of interesting points, but the major limitation of this study does affect the interpretation of the results. The limitation is that COVID-19 infection is self-reported, so those too ill to reply or who have died from it are not represented in the results. That would explain why those younger 50 are not overly represented in the study sample, but also have an increased risk of infection compared to those older than 50 who could answer (as they’d be less likely to be infected). In addition, it would explain why individuals in jobs that were previously shown to be high risk were not found to be high risk here.

This is a significant methodological flaw and limits the ability of the authors to answer their research question accurately

6. PLOS authors have the option to publish the peer review history of their article (what does this mean?). If published, this will include your full peer review and any attached files.

**Do you want your identity to be public for this peer review?** For information about this choice, including consent withdrawal, please see our Privacy Policy.

Reviewer #1: **Yes: **Reza Shahriarirad

Reviewer #2: No

---

## [Decision Letter · Decision Letter 1]

16 Sep 2022

Demographic and professional risk factors of SARS-Cov-2 infections among physicians in low- and middle-income settings; findings from a representative survey in two Brazilian states.

PGPH-D-22-00875R1

Dear Dr Russo,

We are pleased to inform you that your manuscript 'Demographic and professional risk factors of SARS-Cov-2 infections among physicians in low- and middle-income settings; findings from a representative survey in two Brazilian states.' has been provisionally accepted for publication in PLOS Global Public Health.

Best regards,

Julio Croda, Ph.D, M.D.

Academic Editor

Reviewer Comments (if any, and for reference):

Reviewer's Responses to Questions

**Comments to the Author**

1. If the authors have adequately addressed your comments raised in a previous round of review and you feel that this manuscript is now acceptable for publication, you may indicate that here to bypass the “Comments to the Author” section, enter your conflict of interest statement in the “Confidential to Editor” section, and submit your "Accept" recommendation.

Reviewer #1: All comments have been addressed

Reviewer #2: All comments have been addressed

2. Does this manuscript meet PLOS Global Public Health’s publication criteria? Is the manuscript technically sound, and do the data support the conclusions? The manuscript must describe methodologically and ethically rigorous research with conclusions that are appropriately drawn based on the data presented.

Reviewer #1: Yes

Reviewer #2: Yes

3. Has the statistical analysis been performed appropriately and rigorously?

Reviewer #1: Yes

Reviewer #2: Yes

4. Have the authors made all data underlying the findings in their manuscript fully available (please refer to the Data Availability Statement at the start of the manuscript PDF file)?

Reviewer #1: Yes

Reviewer #2: No

5. Is the manuscript presented in an intelligible fashion and written in standard English?

Reviewer #1: Yes

Reviewer #2: Yes

6. Review Comments to the Author

Reviewer #1: The authors have done a satisfactory job in addressing all my comments and I believe that the quality of the manuscript has increased and is ready for publication.

Reviewer #2: Thank you for addressing both reviewer comments. I see your article is in line with existing literature, and your findings are also in line with similar studies. There are still methodological flaws with your data collection that make it impossible claim that "our work appears to confirm that younger doctors in LMICs may be more at risk of infection than their counterparts in HICs". But I congratulate the author for tempering the discussion from the previous review.

I would encourage the authors to share all their data so that their analysis can be checked, or present all their data analysis (e.g., univariate analysis) so that readers can judge for themselves what confounding factors and effect modifiers may be.

7. PLOS authors have the option to publish the peer review history of their article (what does this mean?). If published, this will include your full peer review and any attached files.

**Do you want your identity to be public for this peer review?** For information about this choice, including consent withdrawal, please see our Privacy Policy.

Reviewer #1: **Yes: **Reza Shahriarirad

Reviewer #2: No
